# The Spurious Prospective Associations Model (SPAM): Explaining longitudinal associations due to statistical artifacts

Kimmo Sorjonen[1]*, Bo Melin[1], Gustav Nilsonne[1,2]

1 Department of Clinical Neuroscience, Karolinska Institutet, Stockholm, Sweden, 2 Department of Psychology, Stockholm University, Stockholm, Sweden

* kimmo.sorjonen@ki.se

## Abstract

Analysis of longitudinal data often relies on models which can be prone to statistical artifacts. We have previously shown that several published prospective associations can be explained by a combination of a general association between constructs, imperfect measurement reliability, and regression to the mean. Here, we formalize our analysis of this type of statistical artifact and introduce the Spurious Prospective Associations Model (SPAM). We show that the SPAM performs better than adjusted cross-lagged effects models to explain several observed prospective associations, including new examples involving loneliness and social anxiety and resilience and depressive symptoms, without assuming any true increasing or decreasing effects between constructs over time. Moreover, unlike the models we challenge, the SPAM is consistent with seemingly paradoxical findings indicating simultaneous increasing and decreasing effects between constructs. We conclude that the SPAM agrees well with observed data and is better supported than competing adjusted cross-lagged effects models in the cases investigated here.

## Introduction

One approach to questions of causality in social science is to use longitudinal data. When a variable observed at an earlier time predicts another variable at a later time, this is generally thought to support a prospective relationship. However, analysis of longitudinal data often relies on models which can be prone to statistical artifacts. In a series of recent papers, we have challenged models claiming prospective effects between various constructs (Table 1). In all these cases, the challenged models have purportedly been supported by meta-analyses. A common feature of these meta-analyses is that they extracted zero-order correlations from the included studies and used Equation 1 [1] to estimate prospective effects between constructs while adjusting for a prior value on the outcome. However, it is well known that such

**Data availability statement:** Data, a list of studies included in the meta-analyses, forest plots, and analytic script are available at the Open Science Framework at https://osf.io/yuzq6/.

**Funding:** The author(s) received no specific funding for this work.

**Competing interests:** The authors have declared that no competing interests exist.

**Table 1. Meta-analytic findings that have been challenged and may be explained by the SPAM.**

| Prospective effect | Sign | Model | Reference | Challenger |
|---|---|---|---|---|
| SE→Dep | – | Vulnerability | Sowislo and Orth [7] | Sorjonen et al. [8] |
| Dep→SE | – | Scar | Sowislo and Orth [7] | Sorjonen et al. [8] |
| SE→QR | + | Risk regulation | Harris and Orth [9] | Sorjonen et al. [10] |
| QR→SE | + | Sociometer | Harris and Orth [9] | Sorjonen et al. [10] |
| SS→PTSD | – | Social causation | Wang et al. [11] | Sorjonen and Melin [12] |
| PTSD→SS | – | Social selection | Wang et al. [11] | Sorjonen and Melin [12] |
| ASC→AA | + | Self-enhancement | Wu et al. [13] | Sorjonen et al. [14] |
| AA→ASC | + | Skill development | Wu et al. [13] | Sorjonen et al. [14] |

SE = self-esteem; Dep = depression; QR = quality of social relations; SS = social support;

PTSD = post-traumatic stress disorder; ASC = academic self-concept; AA = academic achievement

adjusted prospective effects may be spurious due to correlations with residuals and regression to the mean [2–6].

As an example, let us assume that teenagers tend to sleep more on weekends compared with elderly people. If a teenager and an elderly person sleep equally long on a particular weekend, we may assume that the teenager has slept uncharacteristically little, i.e., experienced a negative residual, and/or that the elderly person has slept unusually much, i.e., experienced a positive residual. However, as residuals tend to regress toward a mean value of zero, we should expect a more positive change in amount of sleep to the following weekend for the teenager compared with the elderly person. On group level, we should therefore expect the dichotomous variable "being a teenager vs. an elderly person" to have a positive effect on change in amount of sleep to the coming weekend if we include amount of sleep the present weekend as a covariate in the model, even if no group-level change in amount of sleep has taken place. As regression to the mean is independent of the direction of time, we should also expect "being a teenager vs. an elderly person" to have a positive effect on length of sleep the previous weekend if we adjust for length of sleep the present weekend. Hence, both a hypothesis of a true increasing (decreasing) effect and a hypothesis of a spurious prospective association predict the effect of $X_1$ on $Y_2$ while adjusting for $Y_1$ to be positive (negative) (Table 2, row 1).

$$E(\beta_{x1,y2.y1}) = \frac{r_{x1,y2} - r_{x1,y1}r_{y1,y2}}{1 - r_{x1,y1}^2}$$

Equation 1

In our previous papers challenging these models, we have argued that if a positive effect of $X_1$ on $Y_2$ while adjusting for $Y_1$ is claimed to indicate an increasing effect of $X_1$ on subsequent change in Y, then the effect of $X_1$ on $Y_1$ while adjusting for $Y_2$ should be negative (Table 2, row 2). Conversely, if a negative effect of $X_1$ on $Y_2$ while adjusting for $Y_1$ is claimed to indicate a decreasing effect of $X_1$ on subsequent change in Y, then the effect of $X_1$ on $Y_1$ while adjusting for $Y_2$ should be positive. Among individuals with the same value on $Y_2$, those with the lowest value on $Y_1$ have experienced the largest increase in Y between the measurements. However, if the

**Table 2. Sign of regression effects predicted by a hypothesis of a true increasing or decreasing effect and a hypothesis of spurious associations.**

| Effect | Increasing effect | | Decreasing effect | |
|---|---|---|---|---|
| | True | Spurious | True | Spurious |
| 1. $\beta_{x1,y2,y1}$ | positive | positive | negative | negative |
| 2. $\beta_{x1,y1,y2}$ | negative | positive | positive | negative |
| 3. $\beta_{x1,y2-y1}$ | positive | zero or negative | negative | zero or positive |
| 4. $\beta_{x2-x1,y2-y1}$ | positive | zero or positive | negative | zero or negative |

Note: In column 1, the variables are given in the order predictor, outcome, and covariate.

prospective effect of $X_1$ on $Y_2$ while adjusting for $Y_1$ is spurious, we can expect the effect of $X_1$ on $Y_1$ while adjusting for $Y_2$ to have the same sign. In accordance with a hypothesis of spuriousness, but contrary to the challenged models, in our earlier papers on this topic we have found the effect of $X_1$ on $Y_1$ while adjusting for $Y_2$ to have the same sign as the effect of $X_1$ on $Y_2$ while adjusting for $Y_1$.

As indicated by Equation 1, adjusted regression effects are functions of correlations. In Equation 1, the denominator will be positive, except in an unlikely situation with a perfect correlation between $X_1$ and $Y_1$. Hence, the sign of the regression effect is decided by the numerator. We would see a positive effect of $X_1$ on $Y_2$ when adjusting for $Y_1$, suggesting an increasing effect of $X_1$ on Y, if $r_{x1,y2} > r_{x1,y1} \times r_{y1,y2}$ and a paradoxical positive effect of $X_1$ on $Y_1$ when adjusting for $Y_2$, suggesting a decreasing effect of $X_1$ on Y, if $r_{x1,y1} > r_{x1,y2} \times r_{y1,y2}$. Both of these outcomes may happen, e.g., if Y is measured with low reliability, resulting in a low value on $r_{y1,y2}$. This means that a positive (or negative) and statistically significant cross-lagged effect of $X_1$ on $Y_2$ when adjusting for $Y_1$ may be due to imperfect reliability in the measurement of Y rather than due to causality.

The expected standardized non-adjusted effect of $X_1$ on change in Y is given by Equation 2 [15]. With a true increasing effect, this effect is expected to be positive (Table 2, row 3). According to Equation 2, this will be the case if the correlation between $X_1$ and $Y_2$ is more positive than the correlation between $X_1$ and $Y_1$. This would mean a strengthening of the correlation between $X_1$ and Y with time, something that should be expected if $X_1$ has a true increasing effect on Y. With time, individuals' degree of Y would more and more match their value on $X_1$. Contrarily, if $X_1$ has no true increasing effect on Y, the effect of $X_1$ on the $Y_2$-$Y_1$ difference can be expected to be either close to zero, which would mean that the correlation between $X_1$ and $Y_2$ and between $X_1$ and $Y_1$ are approximately equally strong, or negative. The latter case is expected if both X and Y are affected by some common state-factor, as this would tend to result in a stronger correlation between X and Y if they are measured close in time. In accordance with a hypothesis of spuriousness, but contrary to the challenged models, in our earlier papers we did not find the effect of $X_1$ on the $Y_2$-$Y_1$ difference to have the same sign as the effect of $X_1$ on $Y_2$ while adjusting for $Y_1$.

$$E(\beta_{x1,y2-y1}) = \frac{r_{x1,y2} - r_{x1,y1}}{\sqrt{2(1 - r_{y1,y2})}}$$

Equation 2

Going beyond our earlier papers, we here also present Equation 3 (see Appendix for derivation), which can be used to estimate the effect of the $X_2$-$X_1$ difference on the $Y_2$-$Y_1$ difference. We see in the numerator in Equation 3 that if the concurrent correlations ($r_{x1,y1}$ and $r_{x2,y2}$) are more positive than the cross-lagged correlations ($r_{x1,y2}$ and $r_{x2,y1}$), this effect is expected to be positive. Models claiming an increasing effect of X on Y should predict a positive effect of the $X_2$-$X_1$ difference on the $Y_2$-$Y_1$ difference. This would mean that the effect of $X_2$-$X_1$ on $Y_2$-$Y_1$ has the same sign as the effect of $X_1$ on $Y_2$-$Y_1$ (Table 2, row 4). Contrarily, a hypothesis of spuriousness would predict either that both the effect of $X_1$ on $Y_2$-$Y_1$ and

the effect of $X_2$-$X_1$ on $Y_2$-$Y_1$ are close to zero (if concurrent and cross-lagged correlations are approximately equally strong) or that they have opposing signs (if concurrent correlations are stronger than cross-lagged correlations).

$$E(\beta_{x2-x1,y2-y1}) = \frac{r_{x1,y1} + r_{x2,y2} - r_{x1,y2} - r_{x2,y1}}{\sqrt{4(1 - r_{x1,x2})(1 - r_{y1,y2})}}$$

Equation 3

In the present paper, we challenge increasing effects between loneliness and social anxiety which have been the subject of an earlier meta-analysis [16]. Furthermore, in our earlier papers we have hinted at a data-generating model that may explain prospective associations better than the challenged models. Here, we make a more formal, comprehensive, and generalized presentation and name the model: the spurious prospective associations model (SPAM). We also evaluate the usefulness of the SPAM by fitting it on empirical data on resilience and symptoms of depression.

## Method

### Loneliness and social anxiety

Maes et al. [16] extracted zero-order correlations between loneliness and social anxiety symptom severity measured at two occasions ($Lon_1$, $Lon_2$, $SA_1$, and $SA_2$) from ten studies on children, adolescents, and college students (total $N = 3995$, 46% males). However, correlations involving $SA_2$ were lacking from two studies. We refer to Maes et al. for more information on selection of studies, tests of publication bias, etc. Maes et al. used Equation 1 to estimate the standardized regression effect of $Lon_1$ on $SA_2$ while adjusting for $SA_1$, and vice versa, for each included study. We did the same, using correlations in Table 2 in Maes et al. In addition, using Equations 1–3 we estimated the standardized effect of $Lon_1$ on $SA_1$ while adjusting for $SA_2$, the effect of $Lon_1$ on the $SA_2$-$SA_1$ difference, and the effect of the $Lon_2$-$Lon_1$ difference on the $SA_2$-$SA_1$ difference, and vice versa. Loneliness and social anxiety were positively correlated and the predicted sign of these effects according to a hypothesis of a true increasing effect and a hypothesis of spurious prospective associations can be seen in Table 2. We conducted 14 separate random effects meta-analyses (one for each of the effects in Table 2, in both directions, and for six zero-order correlations). Analyses were conducted on Fisher's z-transformed standardized regression effects and correlations, but these were inverted back to non-transformed effects and correlations for the presentation of results.

### The spurious prospective associations model (SPAM)

According to the SPAM (Fig 1), data are generated through the following steps: (1) A trait, e.g., general negativity/positivity, has an effect on two more specific characteristics, e.g., self-esteem and depression (*a1/a2*, assumed to be the same if the two characteristics are positively associated, and equally strong but with opposite signs if they are negatively associated); (2) The two more specific characteristics have an effect on measurements at two occasions (*b* and *c*, respectively); (3) Correlations between measurements of the two characteristics at the same occasion may be accentuated due to influence by a common state-factor (*d*), resulting in stronger concurrent compared with cross-lagged correlations. Although no true increasing or decreasing effects between characteristics are included in the SPAM, it still predicts, due to correlations with residuals and regression to the mean, spurious effects as presented in Table 2.

We evaluated the SPAM through the following steps: (1) From each of the challenged meta-analyses, we extracted the six meta-analytically estimated zero-order correlations between $X_1$, $X_2$, $Y_1$, and $Y_2$ (except for Sowislo and Orth [7], in which case we estimated the correlations from five available datasets, see Sorjonen et al., [8]); (2) We simulated a large ($N = 10,000$) dataset with the same correlations between $X_1$, $X_2$, $Y_1$, and $Y_2$ as extracted in the previous step; (3) We fitted the SPAM (Fig 1) as a structural equation model (SEM) on the simulated data and extracted parameter values (*a1*, *a2*, *b*, *c*, and *d*), model fit (comparative fit index (CFI), Tucker-Lewis index(TLI), and root mean square error of approximation (RMSEA)), as well as predicted correlations between $X_1$, $X_2$, $Y_1$, and $Y_2$; (4) A new large dataset ($N = 10,000$) was simulated, with the same correlations between $X_1$,

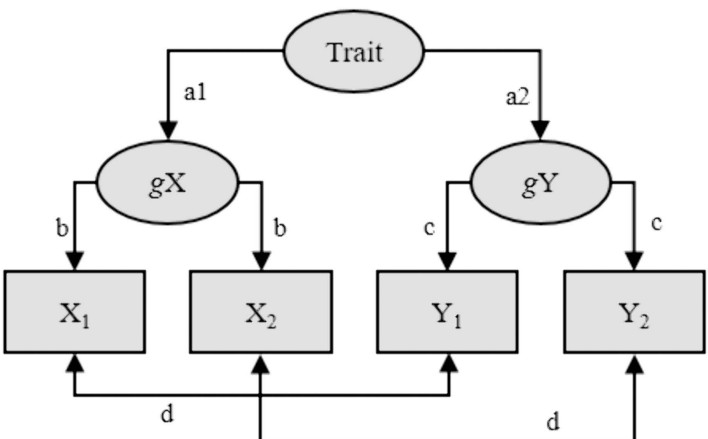

**Fig 1. The spurious prospective associations model (SPAM).** Observations of two characteristics, X and Y, at two occasions are affected by individual general levels of X and Y and by occasion-specific state factors (resulting in correlation *d*). The general levels of X and Y are, in turn, affected by some overarching trait factor. The model predicts a spurious prospective effect of $X_1$ on $Y_2$ when adjusting for $Y_1$, and vice versa. If the effect of the state factors differs from zero (i.e., if $d \neq 0$), the model also predicts a spurious effect of $X_1$ on the $Y_2 - Y_1$ difference, and vice versa, and of the $X_2 - X_1$ difference on the $Y_2 - Y_1$ difference, and vice versa. However, the model predicts these effects to have different signs.

$X_2$, $Y_1$, and $Y_2$ as predicted by the SPAM in the previous step; (5) The following standardized regression effects were calculated in the new dataset: (a) The effect of $X_1$ on $Y_2$ while adjusting for $Y_1$, and vice versa; (b) The effect of $X_1$ on $Y_1$ while adjusting for $Y_2$, and vice versa; (c) The effect of $X_1$ on the $Y_2$-$Y_1$ difference, and vice versa; (d) The effect of the $X_2$-$X_1$ difference on the $Y_2$-$Y_1$ difference, and vice versa (although they are bound to be identical); (6) The effect of $X_1$ on $Y_2$ while adjusting for $Y_1$, and vice versa, calculated in the previous step were compared with the corresponding meta-analytically estimated effects, presented in the challenged studies. In case of high similarity, we may conclude that the meta-analytic findings agree with a data-generating mechanism (i.e., the SPAM) without any increasing or decreasing effects between the two characteristics.

Analyses were conducted with R 4.4.3 statistical software [17], employing the osfr [18], MASS [19], lavaan [20], metafor [21], and foreign [22] packages. Data, a list of studies included in the meta-analyses, forest plots, and analytic script are available at the Open Science Framework at https://osf.io/yuzq6/.

### Resilience and depression

In order to further evaluate the usefulness of the SPAM, we fitted the model on empirical data used and made publicly available (https://osf.io/d3qht/) by Meule et al. [23]. We refer to Meule et al. for more comprehensive information on the study sample, used instruments, study procedures, etc. In short, Meule et al. analyzed data on self-reported resilience and depressive symptoms collected from inpatients with depression ($N=2165$, mean age$=37.9$ years [range: 12−88 years], 66.5% female) at admission and at discharge (average length of stay$=60$ days [range: 1−303 days]). Meule et al. reported a statistically significant negative effect of resilience at admission on depressive symptoms at discharge when adjusting for depressive symptoms at admission ($\beta=-0.09$) and vice versa ($\beta=-0.18$). Meule et al. concluded, for example, that resilience is a pivotal predictor of treatment outcomes and that resilience, therefore, should be promoted in order to prevent mental disorders.

## Results

### Loneliness and social anxiety

The meta-analytically estimated correlations and regression effects between loneliness and social anxiety at two time points are presented in Table 3. All zero-order correlations (rows 1–6) were fairly strong, in an order of decreasing strength

**Table 3. Meta-analytically estimated correlations and adjusted regression effects between loneliness and social anxiety measured at two occasions.**

| Association | K | N | Estimate (95% CI) | Q (df) | I² (95% CI) |
|---|---|---|---|---|---|
| 1. r(L1,L2) | 10 | 3995 | 0.603 (0.463; 0.714) | 373 (9)*** | 97.2 (94.0; 99.2) |
| 2. r(L1,S1) | 10 | 3995 | 0.493 (0.439; 0.544) | 33 (9)*** | 75.4 (43.6; 94.3) |
| 3. r(L1,S2) | 8 | 3703 | 0.379 (0.297; 0.456) | 56 (7)*** | 85.8 (65.5; 96.8) |
| 4. r(L2,S1) | 10 | 3995 | 0.351 (0.300; 0.400) | 24 (9)** | 62.7 (15.5; 88.5) |
| 5. r(L2,S2) | 8 | 3703 | 0.552 (0.491; 0.608) | 60 (7)*** | 82.6 (58.2; 95.1) |
| 6. r(S1,S2) | 8 | 3703 | 0.568 (0.492; 0.636) | 46 (7)*** | 89.3 (73.3; 97.6) |
| 7. β(L1,S2.S1) | 8 | 3703 | 0.123 (0.047; 0.197) | 42 (7)*** | 78.4 (48.4; 94.3) |
| 8. β(L1,S1.S2) | 8 | 3703 | 0.354 (0.275; 0.429) | 36 (7)*** | 84.1 (59.7; 96.4) |
| 9. β(L1,S2-S1) | 8 | 3703 | −0.145 (−0.222; −0.067) | 33 (7)*** | 80.0 (49.9; 95.3) |
| 10. β(L2-L1,S2-S1) | 8 | 3703 | 0.366 (0.267; 0.458) | 71 (7)*** | 90.1 (75.9; 97.7) |
| 11. β(S1,L2.L1) | 10 | 3995 | 0.094 (0.041; 0.146) | 25 (9)** | 56.8 (11.4; 82.0) |
| 12. β(S1,L1.L2) | 10 | 3995 | 0.328 (0.230; 0.419) | 62 (9)*** | 90.0 (77.4; 97.4) |
| 13. β(S1.L2-L1) | 10 | 3995 | −0.153 (−0.204; −0.100) | 20 (9)* | 58.1 (7.56; 90.5) |
| 14. β(S2-S1,L2-L1) | 8 | 3703 | 0.366 (0.267; 0.458) | 71 (7)*** | 90.1 (75.9; 97.7) |

Note: K = number of studies; N = total sample size; Q = Cochran's Q; I² = percentage of variation due to heterogeneity; L = loneliness; S = social anxiety; 1 = time 1; 2 = time 2; the variables are given in the order predictor, outcome, and covariate; *** p < 0.001, ** p < 0.01, * p < 0.5

from autoregressive (rows 1 and 6), to concurrent (rows 2 and 5), and cross-lagged (rows 3 and 4) correlations. The effects of loneliness at time 1 on social anxiety at time 2 while adjusting for social anxiety at time 1 (row 7), and vice versa (row 11), were positive and statistically significant (see Fig 2A for an illustration). These effects have already been shown by Maes et al. [16], who suggested that they indicate a reciprocal increasing effect between loneliness and social anxiety. However, contrary to a hypothesis of true increasing effects, the effects of loneliness at time 1 on social anxiety at time 1 when adjusting for social anxiety at time 2 (row 8), and vice versa (row 12), were also positive. This means that among individuals with the same degree of social anxiety at time 2, those with a high degree of loneliness at time 1 had experienced a higher degree of social anxiety at time 1 and, consequently, a larger decrease in degree of social anxiety between the measurements compared with those with the same degree of social anxiety at time 2 but with a lower degree of loneliness at time

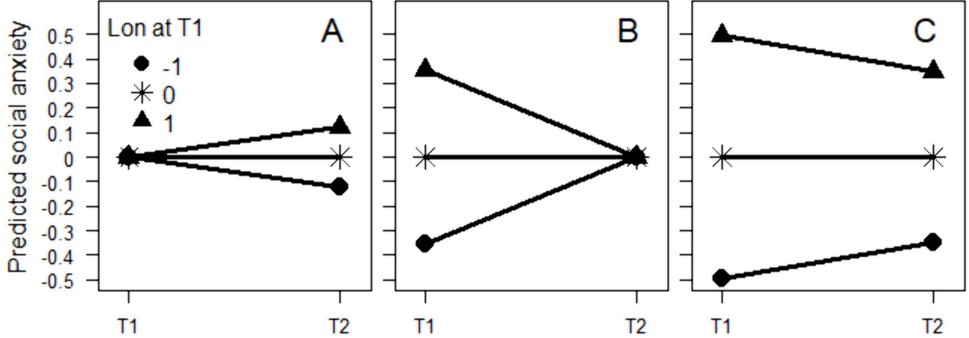

**Fig 2. Predicted anxiety.** Predicted degree of social anxiety symptoms at T1 and T2 separately for those with high (Z = 1), average, and low (Z = −1) degree of loneliness at T1 when conditioning on average social anxiety at T1 (A), when conditioning on average social anxiety at T2 (B), and when not conditioning on social anxiety (C).

1 (Fig 2B). Similarly, among individuals with the same degree of loneliness at time 2, those with a high degree of social anxiety at time 1 had experienced a larger decrease in loneliness between the measurements compared with individuals with low degree of social anxiety at time 1. Furthermore, the effects of loneliness at time 1 on the social anxiety at time 2 – social anxiety at time 1 difference (Table 3, row 9), and vice versa (row 13), were negative (Fig 2C), which can be attributed to stronger concurrent (rows 2 and 5) compared with cross-lagged (rows 3 and 4) correlations. The difference in strength between concurrent and cross-lagged correlations can also account for the positive effect of the loneliness at time 2 – loneliness at time 1 difference on the social anxiety at time 2 – social anxiety at time 1 difference (row 10), and vice versa (row 14). In summary, the sign of the effects agreed better with a hypothesis of spurious prospective associations than with a hypothesis of true increasing effects (Table 2). There was heterogeneity, as indicated by a significant Cochran's $Q$ test and high $I^2$.

## The SPAM

Meta-analytically estimated zero-order correlations between five pairs of variables, measured at two occasions, are presented in Table 4. For each pair (row) the strength of the correlations is in a decreasing order from autoregressive (columns 4 and 9), to concurrent (columns 5 and 8), and cross-lagged (columns 6 and 7) correlations. Stronger concurrent compared with cross-lagged correlations suggest that the variables have been affected by a common state-factor.

Five large ($N = 10,000$) datasets with four variables ($X_1$, $X_2$, $Y_1$, and $Y_2$) with the same zero-order correlations as in Table 4 were simulated and the SPAM (Fig 1) was fitted to the data as a SEM. Although the results did not perfectly reproduce the correlations in Table 4, in which case the $\chi^2$-value of the model (Table 5, column 8) would have been zero, the SPAM showed excellent fit as indicated by high CFI (column 9) and TLI (column 10) values and a low RMSEA (column 11) value. The high $a1$ (column 3) and $a2$ (column 4) SPAM-parameters suggest that the constructs (e.g., self-esteem and depression in row 1 in Table 5) were strongly influenced by a common trait-factor. Furthermore, high $b$ (column 5) and $c$ (column 6) parameters indicate a high stability of the constructs. The $d$ (column 7) parameter indicates a stronger influence by a common state-factor on (self-rated) self-esteem and depression (row 1) and loneliness and social anxiety (row 5) and a weaker influence on self-esteem and quality of social relations (row 2), social support and PTSD (row 3), and academic self-concept and achievement (row 4).

Five new large ($N = 10,000$) datasets with four variables ($X_1$, $X_2$, $Y_1$, and $Y_2$) were simulated, now with the zero-order correlations between variables predicted by the fitted SPAM. In the presentation in Fig 3, variables take turns being the X and the Y variable. For example, on row 1 in Fig 3, X = self-esteem and Y = depression while on row 2, X = depression and Y = self-esteem, although these effects have been estimated in the same simulated dataset. Similarly, effects on rows 3 and 4 in Fig 3 have been estimated in the same simulated dataset, etc. In each dataset we estimated:

**Table 4. Meta-analytically estimated zero-order correlations.**

| 1.Reference | 2.X | 3.Y | 4.$r_{x1,x2}$ | 5.$r_{x1,y1}$ | 6.$r_{x1,y2}$ | 7.$r_{x2,y1}$ | 8.$r_{x2,y2}$ | 9.$r_{y1,y2}$ |
|---|---|---|---|---|---|---|---|---|
| Sorjonen et al. [8] | SE | Dep | 0.70[1] | −0.51[1] | −0.39[1] | −0.41[1] | −0.53[1] | 0.53[1] |
| Sorjonen et al. [10] | SE | QR | 0.57 | 0.28 | 0.26 | 0.24 | 0.34 | 0.60 |
| Wang et al. [11] | SS | PTSD | 0.55 | −0.26 | −0.24 | −0.22 | −0.26[2] | 0.63 |
| Sorjonen et al. [14] | ASC | AA | 0.54 | 0.37 | 0.34 | 0.34 | 0.41 | 0.70 |
| The present study | Lon | SA | 0.60 | 0.49 | 0.38 | 0.35 | 0.55 | 0.57 |

[1] Not reported in the reference, calculated for the present study as the mean across four waves of measurement in five available datasets;

[2] Not reported in the reference, assumed to be the same as $r_{x1,y1.}$ SE = self-esteem; Dep = depression; QR = quality of social relations; SS = social support; PTSD = post-traumatic stress disorder; ASC = academic self-concept; AA = academic achievement; Lon = loneliness; SA = social anxiety.

**Table 5. Estimated parameter values and model fit when the SPAM (Fig 1) was fitted on datasets. (N = 10,000) with the same zero-order correlations as has been meta-analytically estimated (Table 4).**

| 1.X | 2.Y | 3.a1 | 4.a2 | 5.b | 6.c | 7.d | 8.$\chi^2$ | 9.CFI | 10.TLI | 11.RMSEA |
|---|---|---|---|---|---|---|---|---|---|---|
| 1.SE | Dep | 0.809 | −0.809 | 0.838 | 0.730 | −0.123 | 12.59 | 1.000 | 1.000 | 0.010 |
| 2.SE | QR | 0.655 | 0.655 | 0.755 | 0.775 | 0.060 | 30.14 | 0.998 | 0.998 | 0.020 |
| 3.SS | PTSD | 0.625 | −0.625 | 0.742 | 0.794 | −0.030 | 3.18 | 1.000 | 1.000 | 0.000 |
| 4.ASC | AA | 0.744 | 0.744 | 0.735 | 0.837 | 0.050 | 13.01 | 0.999 | 0.999 | 0.011 |
| 5.Lon | SA | 0.790 | 0.790 | 0.777 | 0.755 | 0.158 | 28.14 | 0.998 | 0.998 | 0.019 |

SE = self-esteem; Dep = depression; QR = quality of social relations; SS = social support; PTSD = post-traumatic

stress disorder; ASC = academic self-concept; AA = academic achievement; Lon = loneliness; SA = social anxiety

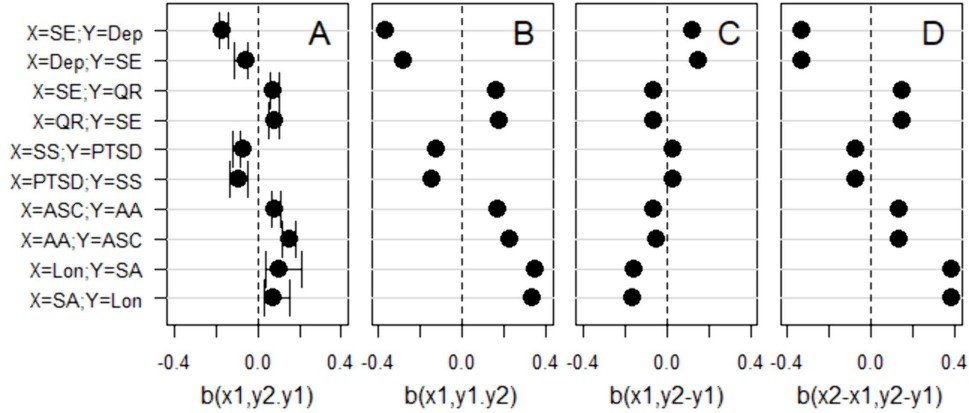

**Fig 3. Estimated regression effects.** Estimated regression effect of $X_1$ on $Y_2$ when adjusting for $Y_1$ (panel A), of $X_1$ on $Y_1$ when adjusting for $Y_2$ (B), of $X_1$ on the $Y_2$-$Y_1$ difference (C), and of the $X_2$-$X_1$ difference on the $Y_2$-$Y_1$ difference (D), separately for ten different combinations of X and Y variables (rows). Effects in each pair of rows (i.e., 1 and 2; 3 and 4; etc.) were estimated in the same simulated dataset (N = 10,000). The simulated data were generated by the SPAM (Fig 1) with the parameter values presented in Table 5. For the effects of $X_1$ on $Y_2$ while adjusting for $Y_1$ (panel A), we have added 95% confidence intervals of the meta-analytically estimated effect, extracted from the challenged studies. SE = self-esteem; Dep = depression; QR = quality of social relations; SS = social support; PTSD = post-traumatic stress disorder; ASC = academic self-concept; AA = academic achievement; Lon = loneliness; SA = social anxiety.

(1) The effect of $X_1$ on $Y_2$ while adjusting for $Y_1$ (Fig 3A). The 95% confidence intervals of the corresponding effects, extracted from the challenged studies, are included in Fig 3A. Effects predicted by the SPAM matched the extracted confidence intervals very closely;

(2) The effect of $X_1$ on $Y_1$ while adjusting for $Y_2$ (Fig 3B). In each of the ten cases (rows in Fig 3), this effect had the same sign as the effect of $X_1$ on $Y_2$ while adjusting for $Y_1$ (Fig 3A). This means that if one of these effects indicated an increasing effect of $X_1$ on Y, the other effect indicated, paradoxically, a decreasing effect of $X_1$ on Y;

(3) The effect of $X_1$ on the $Y_2$-$Y_1$ difference (Fig 3C). In each of the ten cases (rows), this effect had the opposite sign as the effect of $X_1$ on $Y_2$ while adjusting for $Y_1$ (Fig 3A). This means, again, that if one of these effects indicated an increasing effect of $X_1$ on Y, the other effect indicated, paradoxically, a decreasing effect of $X_1$ on Y. The effect of $X_1$ on the $Y_2$-$Y_1$ difference (Fig 3C), can be attributed (Equation 2) to the difference in strength between concurrent and cross-lagged correlations, i.e., to the influence by occasion-specific state factors;

(4) The effect of the $X_2$-$X_1$ difference on the $Y_2$-$Y_1$ difference (Fig 3D). These effects are identical on each pair of rows, i.e., in the same dataset. In each of the 5×2 = 10 cases (rows), this effect had the opposite sign as the effect of $X_1$ on the $Y_2$-$Y_1$ difference (Fig 3C). This means that if $X_1$ had an increasing effect on Y, the $X_2$-$X_1$ difference had, paradoxically, a decreasing effect on Y. The effect of the $X_2$-$X_1$ difference on the $Y_2$-$Y_1$ difference (Fig 3D), can be attributed (Equation 3) to the difference in strength between concurrent and cross-lagged correlations, i.e., to the influence by occasion-specific state factors.

In summary, the SPAM accurately reproduced the effects of $X_1$ on $Y_2$ while adjusting for $Y_1$, and vice versa, meta-analytically estimated in the challenged studies. However, the combination of signs of estimated regression effects suggests (compare effects in Fig 3 with predictions in Table 2) that they were spurious rather than truly increasing or decreasing.

## Resilience and depression

In agreement with findings by Meule et al. [23], our reanalyses of the same empirical data indicated a negative effect of resilience at admission on depressive symptoms at discharge when adjusting for depressive symptoms at admission (β = −0.095 [−0.144; −0.046], $p < 0.001$). This could be seen to suggest, as Meule et al. concluded, that resilience has a decreasing/protective prospective effect on mental disorders. However, contradicting this conclusion, a negative effect of resilience at admission on depressive symptoms at admission when adjusting for depressive symptoms at discharge (β = −0.209 [−0.256; −0.162], $p < 0.001$) and a positive effect on the depressive symptoms at discharge – depressive symptoms at admission difference score (β = 0.080 [0.022; 0.139], $p = 0.007$) suggested, paradoxically, an increasing prospective effect of resilience on mental disorders. As argued and shown above, these paradoxical effects would be expected if the data were generated by the SPAM, which had a good fit (Fig 4). It can be noted that parameter values in Fig 4 are similar to the corresponding values when fitting the SPAM on data simulated to resemble meta-analytic data on associations between self-esteem and depression (row 1 in Table 5).

## Discussion

A first objective of the present study was to evaluate the proposition by Maes et al. [16] that loneliness and social anxiety symptoms have a reciprocal increasing prospective effect on each other. Our analyses indicated that depending on the

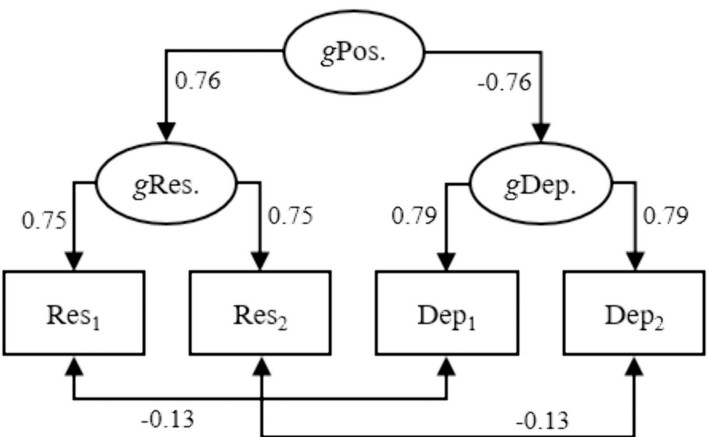

**Fig 4. The SPAM fitted on empirical data.** The SPAM fitted on empirical data on resilience and depressive symptoms at admission and at discharge among psychiatric inpatients (N = 2165). All effects (standardized) were highly statistically significant ($p < 0.001$) and the model had good fit ($\chi^2 = 26.0$, $DF = 6$, $p < 0.001$, CFI = 0.988, TLI = 0.988, RMSEA = 0.039 [90% CI: 0.025; 0.055]). Note: gPos. = general positivity; gRes. = general resilience; gDep. = general depressivity; $Res_1$, $Res_2$, $Dep_1$, $Dep_2$ = resilience and depression scores at admission and at discharge, respectively.

chosen model one could claim, simultaneously, that loneliness and social anxiety have both an increasing and a decreasing effect on each other's subsequent change. Due to these incongruent findings, we propose that prospective associations between loneliness and social anxiety may be spurious rather than truly increasing.

A second objective was to introduce and evaluate the spurious prospective associations model (SPAM). The SPAM excellently reproduced meta-analytically estimated zero-order correlations between two constructs measured at two occasions (extracted from several challenged studies), as indicated by excellent model fits. The SPAM also did a good job at reproducing associations between scores on resilience and depressive symptoms in a two-wave empirical dataset collected from psychiatric inpatients. Furthermore, the SPAM well reproduced adjusted prospective effects between self-esteem and depression [7], self-esteem and quality of social relations [9], social support and PTSD [11], academic self-concept and achievement [13], and loneliness and social anxiety [16]. However, according to the SPAM, prospective effects are spurious due to a combination of a general association between the constructs (which could be due to a confounding influence by some overarching trait), less than perfect reliability in the measurements, and regression to the mean. Unlike the challenged models, the SPAM is consistent with results indicating simultaneous increasing and decreasing effects between constructs. Consequently, we propose the SPAM as a viable and generic alternative to the challenged models, i.e., (1) the vulnerability and scar models of the association between self-esteem and depression; (2) the risk regulation and sociometer models of the association between self-esteem and quality of social relations; (3) the social causation and social selection models of the association between social support and PTSD; (4) the self-enhancement and skill development models of the association between self-concepts and achievement (however, see Limitations below); (5) unnamed models suggesting increasing effects between loneliness and social anxiety.

The present study carries clinical relevance. The SPAM warns against assuming that a prospective cross-lagged effect of initial X (e.g., self-rated resilience) on subsequent Y (e.g., self-rated depressive symptoms) when adjusting for initial Y indicates a causal effect. Hence, it is not certain that measures targeting X will result in changes in Y. Therefore, if improving levels of Y is a prioritized goal, it is probably advisable to require more evidence than cross-lagged effects in observational (i.e., non-experimental) data before investing limited resources into changing X.

In a recent critique of the approach we have taken in earlier papers and here, Lucas et al. [24] claim that observations of an unexpected sign in an estimated coefficient may not indicate that results are spurious. For example, they write that a negative effect of $X_1$ on the $Y_2$-$Y_1$ difference in a longitudinal model is compatible with a positive causal effect of $X_1$ on $Y_2$. However, in a response we argue that Lucas et al. appear to assume that causality is best viewed as an effect of $X_1$ on the $Y_2$-$Y_1$ difference while adjusting for $Y_1$, which may result in unreasonable conclusions in some situations. We argue that it may often be better to require an unadjusted effect of $X_1$ on the $Y_2$-$Y_1$ difference before claiming causality. Moreover, the critique by Lucas et al. appears, in our opinion, to focus on avoiding type 2-errors (i.e., missing true effects) and to oversee the importance of avoiding type 1-errors (i.e., making false positive claims) [25].

As mentioned above, unreliability of the cross-lagged panel model has been pointed out before [2–6]. The random-intercept cross-lagged panel model (RI-CLPM) is an extension of the traditional cross-lagged panel model, developed to counteract some of these problems. In the RI-CLPM, autoregressive and cross-lagged effects are estimated while adjusting for individuals' stable trait-like levels on the two constructs. In this way, effects are purportedly estimated within individuals, rather than between individuals as in the traditional model [26,27]. It has been argued that within-individual effects are better estimates of causality compared with between-individual effects [26,28]. The RI-CLPM and the SPAM are both structural equation models for longitudinal data. However, two distinguishing characteristics are: (1) The RI-CLPM appears mainly to be used by researchers who wish to unearth causal effects in observational (i.e., non-experimental) data. However, the RI-CLPM cannot do this as it cannot adjust for time-varying confounding [29,30]. This means that the RI-CLPM is susceptible to similar spurious findings as the traditional cross-lagged panel model. The SPAM, on the other hand, has been developed to show that data including cross-lagged effects may often have been generated without any genuine causal effects and that causal conclusions, therefore, are not warranted; (2) The SPAM, at least in its present

initial version, is devised for data from two waves of measurement. The RI-CLPM, on the other hand, requires data from at least three waves of measurement.

## Limitations

The present reanalysis of the meta-analysis by Maes et al. [16] suffered from the same limitations as the original study. For example, only studies on children, adolescents, and college students were included and seven of the ten studies were conducted in the United States. Hence, it is unclear if the present finding, that prospective effects between loneliness and social anxiety appear to be spurious rather than truly increasing, generalizes to other populations and cultural contexts.

In our reanalysis of the meta-analysis by Maes et al. [16], we did not consider possible moderators, such as time between measurements, used instruments, gender composition of the sample, etc. However, it is important to bear in mind that possible moderators were constant across the different analyzed models. Consequently, they cannot explain why, for example, one model indicated an increasing effect of loneliness on subsequent change in social anxiety (Fig 2A) while other models indicated a decreasing effect (Fig 2B and Fig 2C).

The present study was largely based on simulated data, although we also applied the SPAM on empirical data on resilience and depressive symptoms. Simulations may be questioned as the true data generating model/mechanism is not known. However, the same critique can be directed at empirical findings. For example, Sowislo and Orth [7] claimed, in accordance with the vulnerability model, that a meta-analytically estimated negative prospective effect of self-esteem on subsequent depression ratings while adjusting for prior depression ratings suggested a decreasing effect of self-esteem on depression. However, Sowislo and Orth could not know that data had been generated by a mechanism involving a true decreasing effect. As demonstrated in the present study, the prospective effect agrees with a situation where data has been generated by the SPAM without any true decreasing effect, i.e., the prospective effect may be spurious. Although simulations may not prove which data generating mechanism is the correct one, they can indicate which data generating mechanisms are possible. Consequently, we do not claim that the present findings prove, once and for all, that the challenged models are false. We do claim, however, that the present findings show that the challenged models may be false. Moreover, due to the contradictory findings, indicating simultaneous increasing and decreasing effects, we suggest that the SPAM is, at least for now, a better supported alternative to the challenged models.

Although the SPAM suggests that associations between more specific constructs are due to a confounding influence by some overarching trait-factor (Fig 1), this does not always have to be the case. For example, we have argued that an association between academic self-concept (i.e., self-rated academic ability) and academic achievement may be due to a cross-sectional and unidirectional effect of the latter on the former [14,31]. This hypothesis would correspond with a modified SPAM, where the trait-factor is replaced by a unidirectional regression effect. However, the fit of the modified and the original SPAM would be exactly the same and would, consequently, not help to choose between the models. It should be noted that this is a general limitation of SEM:s and not specific for the SPAM.

## Conclusions

Claims about increasing or decreasing effects between constructs, e.g., loneliness and social anxiety, based on cross-lagged effects while adjusting for a prior measurement of the outcome variable, may be false, even if the effects have been identified by meta-analyses. Effects may be spurious due to correlations with residuals and regression to the mean. The spurious prospective associations model (SPAM) can reproduce meta-analytically estimated correlations and prospective regressions effects, without assuming any true increasing or decreasing effects between constructs. Moreover, unlike challenged models, the SPAM agrees with seemingly paradoxical findings indicating simultaneous increasing and decreasing effects between constructs. We conclude that in the cases examined here, the SPAM is better supported than competing models.

## Appendix

The expected standardized crude regression effect of $X_1$ on the $Y_2$-$Y_1$ (both variables standardized) difference is given by e1 (Guilford, 1965).

$$E(\beta_{x1,y2-y1}) = \frac{r_{x1,y2} - r_{x1,y1}}{\sqrt{2(1 - r_{y1,y2})}}$$

e1

We can replace $X_1$ in e1 by $X_2$-$X_1$:

$$E(\beta_{x2-x1,y2-y1}) = \frac{r_{x2-x1,y2} - r_{x2-x1,y1}}{\sqrt{2(1 - r_{y1,y2})}}$$

e2

As zero-order correlations equal standardized crude regression effects, we can estimate the two correlations in the numerator of e2 with e1:

$$E(\beta_{x2-x1,y2-y1}) = \frac{\left[\frac{r_{x2,y2} - r_{x1,y2}}{\sqrt{2(1 - r_{x1,x2})}}\right] - \left[\frac{r_{x2,y1} - r_{x1,y1}}{\sqrt{2(1 - r_{x1,x2})}}\right]}{\sqrt{2(1 - r_{y1,y2})}}$$

e3

As the two expressions within square brackets in e3 have the same denominator, we can move it to the main denominator:

$$E(\beta_{x2-x1,y2-y1}) = \frac{(r_{x2,y2} - r_{x1,y2}) - (r_{x2,y1} - r_{x1,y1})}{\sqrt{2(1 - r_{x1,x2})}\sqrt{2(1 - r_{y1,y2})}}$$

e4

We can remove the brackets in and rearrange the numerator and merge the expressions in the denominator in e4:

$$E(\beta_{x2-x1,y2-y1}) = \frac{r_{x1,y1} + r_{x2,y2} - r_{x1,y2} - r_{x2,y1}}{\sqrt{4(1 - r_{x1,x2})(1 - r_{y1,y2})}}$$

e5

e5 is identical to Equation 3 in the paper.

## Acknowledgments

We thank our colleague Laura Caquelin for checking the code and verifying the results as reported.

## Author contributions

**Conceptualization:** Kimmo Sorjonen, Bo Melin, Gustav Nilsonne.

**Data curation:** Kimmo Sorjonen.

**Formal analysis:** Kimmo Sorjonen.

**Investigation:** Kimmo Sorjonen, Bo Melin, Gustav Nilsonne.

**Methodology:** Kimmo Sorjonen.

**Project administration:** Kimmo Sorjonen.

**Software:** Kimmo Sorjonen.

**Supervision:** Bo Melin, Gustav Nilsonne.

**Validation:** Bo Melin, Gustav Nilsonne.

**Visualization:** Kimmo Sorjonen.

**Writing – original draft:** Kimmo Sorjonen.

**Writing – review & editing:** Kimmo Sorjonen, Bo Melin, Gustav Nilsonne.

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
