## [Decision Letter · Decision Letter 0]

14 Jul 2025

Dear Dr. Sorjonen,

We look forward to receiving your revised manuscript.

Kind regards,

Hikmat Ullah Khan, PhD (Computer Science)

Academic Editor

PLOS ONE

Journal Requirements:

Reviewers' comments:

Reviewer's Responses to Questions

**Comments to the Author**

1. Is the manuscript technically sound, and do the data support the conclusions?

Reviewer #1: Yes

Reviewer #2: Yes

2. Has the statistical analysis been performed appropriately and rigorously?

Reviewer #1: Yes

Reviewer #2: Yes

3. Have the authors made all data underlying the findings in their manuscript fully available?

Reviewer #1: Yes

Reviewer #2: Yes

4. Is the manuscript presented in an intelligible fashion and written in standard English?

Reviewer #1: Yes

Reviewer #2: Yes

Reviewer #1: 1. The biggest issue is that most of the support for SPAM comes from computer simulations. While these are useful, it is more interesting to test more with real-life data from different groups.

2. While the paper is good at showing how SPAM can explain weird patterns, it doesn’t spend much time giving reasons these patterns might happen.

3. There’s a lot of discussion about how different studies have very different results, but not much explanation for why that’s happening. Exploring this could make the conclusions stronger.

Reviewer #2: Comments to the Author

Technical Soundness & Data Support for Conclusions

The manuscript is technically sound, with robust methodology and transparent reporting. The conclusions are well-supported by the data, and the analytic approach is appropriate for the research question. The work meets high standards for technical rigor and data-driven inference)

Statistical Rigor

Assessment:-

Generally acceptable with minor clarifications needed

The manuscript demonstrates a high standard of statistical rigor. The analytic approach is methodologically sound, transparent, and well-justified for the research question. The results are robustly supported by appropriate statistical analyses, and the limitations are candidly addressed. The work meets the expectations for statistical rigor in high-quality empirical research

Methodological Standards

The manuscript meets or exceeds methodological standards for publication in PLOS ONE. The research design, analytic approach, transparency, and reporting are all exemplary. The work is a model of methodological rigor and transparency, and the conclusions are well-supported by the evidence presented. No major methodological concerns are identified.

Conclusion Support

The conclusions presented in the manuscript, "The Spurious Prospective Associations Model (SPAM) Explaining longitudinal associations due to statistical artifacts," are well-supported by the data, analyses, and arguments provided.

The authors clearly demonstrate, both conceptually and empirically, that many previously reported prospective associations in longitudinal psychological research can be explained by statistical artifacts specifically, regression to the mean, imperfect measurement reliability, and general associations between constructs rather than by true causal effects.

Data Repository Compliance

Open Data and Code

All relevant data, analytic scripts, and supporting materials are made available via the Open Science Framework, allowing for full transparency and reproducibility.

Presentation & Language

The manuscript is well organized and communicates complex statistical concepts effectively. The introduction, methods, and results are clearly delineated, and the use of figures and tables enhances comprehension. The writing is accessible to interdisciplinary researchers while maintaining technical precision.

Detailed Feedback

Strengths

Rigorous methodological approach

Transparent and reproducible analysis

Important implications for the interpretation of longitudinal data.

Major Issues

After a thorough review of the manuscript The Spurious Prospective Associations Model (SPAM) Explaining longitudinal associations due to statistical artifacts (PONE-D-24-52718), I find the work to be methodologically rigorous and clearly presented.

However, for a high-impact journal such as PLOS ONE, several major issues should be addressed to strengthen the manuscript and its implications for the field

Minor Issues

The model does not account for measurement error via latent variables, which could strengthen its robustness

Further comparison to models like the Random Intercept Cross-Lagged Panel Model (RI-CLPM) could enhance the discussion of alternatives

Suggestions

Model Comparison:-

Consider including a brief discussion of how SPAM compares with other modern alternatives such as RI-CLPM or Latent Change Score models

External Validation:-

While the simulations are compelling, a worked example using real raw data (not just correlations) could strengthen the argument

Practical Implications:-

Consider providing examples of how SPAM might inform decisions in clinical, social, or policy interventions..

**Do you want your identity to be public for this peer review?** For information about this choice, including consent withdrawal, please see our Privacy Policy

Reviewer #1: No

Reviewer #2: No

---

## [Author Response · Author response to Decision Letter 1]

16 Jul 2025

Reviewers' comments:

Reviewer's Responses to Questions

Comments to the Author

1. Is the manuscript technically sound, and do the data support the conclusions?

Reviewer #1: Yes

Reviewer #2: Yes

2. Has the statistical analysis been performed appropriately and rigorously?

Reviewer #1: Yes

Reviewer #2: Yes

3. Have the authors made all data underlying the findings in their manuscript fully available?

Reviewer #1: Yes

Reviewer #2: Yes

4. Is the manuscript presented in an intelligible fashion and written in standard English?

Reviewer #1: Yes

Reviewer #2: Yes

5. Review Comments to the Author

Reviewer #1:

1. The biggest issue is that most of the support for SPAM comes from computer simulations. While these are useful, it is more interesting to test more with real-life data from different groups.

Response: An advantage of simulations is that they show what effects can be expected when data are generated through a known/defined model. In the present study we show that all of the challenged findings are compatible with data being generated by the SPAM, i.e., without any genuine effects between X and Y. We have now added analyses of empirical data on resilience and depressive symptoms. See under the heading “Resilience and depression” (lines 194 and 322).

2. While the paper is good at showing how SPAM can explain weird patterns, it doesn’t spend much time giving reasons these patterns might happen.

Response: We assume that you with “weird patterns” refer to the fact that the same data can be used to support both increasing and decreasing effects of X on Y. As we argue in the paper, this may happen if data are generated by the SPAM. We have added the following elaboration (lines 77-86):

As indicated by Equation 1, adjusted regression effects are functions of correlations. In Equation 1, the denominator will be positive, except in an unlikely situation with a perfect correlation between X1 and Y1. Hence, the sign of the regression effect is decided by the numerator. We would see a positive effect of X1 on Y2 when adjusting for Y1, suggesting an increasing effect of X1 on Y, if rx1,y2 > rx1,y1 × ry1,y2 and a paradoxical positive effect of X1 on Y1 when adjusting for Y2, suggesting a decreasing effect of X1 on Y, if rx1,y1 > rx1,y2 × ry1,y2. Both of these outcomes may happen, e.g., if Y is measured with low reliability, resulting in a low value on ry1,y2. This means that a positive (or negative) and statistically significant cross-lagged effect of X1 on Y2 when adjusting for Y1 may be due to imperfect reliability in the measurement of Y rather than due to causality.

3. There’s a lot of discussion about how different studies have very different results, but not much explanation for why that’s happening. Exploring this could make the conclusions stronger.

Response: One reason why different studies have different results might be that they study different constructs. We see no reason to expect the same results in, for example, a study of the association between self-esteem and depression and a study of the association between academic self-concept and achievement.

Reviewer #2:

Comments to the Author

Technical Soundness & Data Support for Conclusions

The manuscript is technically sound, with robust methodology and transparent reporting. The conclusions are well-supported by the data, and the analytic approach is appropriate for the research question. The work meets high standards for technical rigor and data-driven inference)

Statistical Rigor

Assessment:-

• Generally acceptable with minor clarifications needed

The manuscript demonstrates a high standard of statistical rigor. The analytic approach is methodologically sound, transparent, and well-justified for the research question. The results are robustly supported by appropriate statistical analyses, and the limitations are candidly addressed. The work meets the expectations for statistical rigor in high-quality empirical research

Methodological Standards

The manuscript meets or exceeds methodological standards for publication in PLOS ONE. The research design, analytic approach, transparency, and reporting are all exemplary. The work is a model of methodological rigor and transparency, and the conclusions are well-supported by the evidence presented. No major methodological concerns are identified.

Conclusion Support

The conclusions presented in the manuscript, "The Spurious Prospective Associations Model (SPAM) Explaining longitudinal associations due to statistical artifacts," are well-supported by the data, analyses, and arguments provided.

The authors clearly demonstrate, both conceptually and empirically, that many previously reported prospective associations in longitudinal psychological research can be explained by statistical artifacts specifically, regression to the mean, imperfect measurement reliability, and general associations between constructs rather than by true causal effects.

Data Repository Compliance

Open Data and Code

All relevant data, analytic scripts, and supporting materials are made available via the Open Science Framework, allowing for full transparency and reproducibility.

Presentation & Language

The manuscript is well organized and communicates complex statistical concepts effectively. The introduction, methods, and results are clearly delineated, and the use of figures and tables enhances comprehension. The writing is accessible to interdisciplinary researchers while maintaining technical precision.

Detailed Feedback

Strengths

• Rigorous methodological approach

• Transparent and reproducible analysis

• Important implications for the interpretation of longitudinal data.

Major Issues

After a thorough review of the manuscript The Spurious Prospective Associations Model (SPAM) Explaining longitudinal associations due to statistical artifacts (PONE-D-24-52718), I find the work to be methodologically rigorous and clearly presented.

However, for a high-impact journal such as PLOS ONE, several major issues should be addressed to strengthen the manuscript and its implications for the field

Minor Issues

The model does not account for measurement error via latent variables, which could strengthen its robustness

Response: With data from two waves of measurement, it is impossible to distinguish between genuine change and random fluctuations around individuals’ true scores on the construct. The traditional cross-lagged panel model assumes that observed changes are genuine. However, we show that a model (the SPAM) where changes are assumed to be random fluctuations will usually fit the data just as well. However, the SPAM, similarly to all other models, is not able to unearth true causality from to waves of observational data.

Further comparison to models like the Random Intercept Cross-Lagged Panel Model (RI-CLPM) could enhance the discussion of alternatives

Response: See below.

Suggestions

Model Comparison:-

• Consider including a brief discussion of how SPAM compares with other modern alternatives such as RI-CLPM or Latent Change Score models

Response: We have added the following (lines 395-412):

As mentioned above, unreliability of the cross-lagged panel model has been pointed out before [2–6]. The random-intercept cross-lagged panel model (RI-CLPM) is an extension of the traditional cross-lagged panel model, developed to counteract some of these problems. In the RI-CLPM, autoregressive and cross-lagged effects are estimated while adjusting for individuals’ stable trait-like levels on the two constructs. In this way, effects are purportedly estimated within individuals, rather than between individuals as in the traditional model [26,27]. It has been argued that within-individual effects are better estimates of causality compared with between-individual effects [26,28]. The RI-CLPM and the SPAM are both structural equation models for longitudinal data. However, two distinguishing characteristics are: (1) The RI-CLPM appears mainly to be used by researchers who wish to unearth causal effects in observational (i.e., non-experimental) data. However, the RI-CLPM cannot do this as it cannot adjust for time-varying confounding [29,30]. This means that the RI-CLPM is susceptible to similar spurious findings as the traditional cross-lagged panel model. The SPAM, on the other hand, has been developed to show that data including cross-lagged effects may often have been generated without any genuine causal effects and that causal conclusions, therefore, are not warranted; (2) The SPAM, at least in its present initial version, is devised for data from two waves of measurement. The RI-CLPM, on the other hand, requires data from at least three waves of measurement.

External Validation:-

• While the simulations are compelling, a worked example using real raw data (not just correlations) could strengthen the argument

Response: An advantage of simulations is that they show what effects can be expected when data are generated through a known/defined model. In the present study we show that all of the challenged findings are compatible with data being generated by the SPAM, i.e., without any genuine effects between X and Y. We have now added analyses of empirical data on resilience and depressive symptoms. See under the heading “Resilience and depression” (lines 204 and 332).

Practical Implications:-

• Consider providing examples of how SPAM might inform decisions in clinical, social, or policy interventions..

Response: We have added the following (lines 377-383):

The present study carries clinical relevance. The SPAM warns against assuming that a prospective cross-lagged effect of initial X (e.g., self-rated resilience) on subsequent Y (e.g., self-rated depressive symptoms) when adjusting for initial Y indicates a causal effect. Hence, it is not certain that measures targeting X will result in changes in Y. Therefore, if improving levels of Y is a prioritized goal, it is probably advisable to require more evidence than cross-lagged effects in observational (i.e., non-experimental) data before investing limited resources into changing X.

---

## [Decision Letter · Decision Letter 1]

19 Aug 2025

The Spurious Prospective Associations Model (SPAM): Explaining longitudinal associations due to statistical artifacts

PONE-D-24-52718R1

Dear Dr. Sorjonen,

We’re pleased to inform you that your manuscript has been judged scientifically suitable for publication and will be formally accepted for publication once it meets all outstanding technical requirements.

Kind regards,

Hikmat Ullah Khan, PhD (Computer Science)

Academic Editor

PLOS ONE

Additional Editor Comments (optional):

Reviewers' comments:

Reviewer's Responses to Questions

**Comments to the Author**

Reviewer #1: All comments have been addressed

2. Is the manuscript technically sound, and do the data support the conclusions?

Reviewer #1: Yes

3. Has the statistical analysis been performed appropriately and rigorously?

Reviewer #1: Yes

4. Have the authors made all data underlying the findings in their manuscript fully available?

Reviewer #1: Yes

5. Is the manuscript presented in an intelligible fashion and written in standard English?

Reviewer #1: Yes

Reviewer #1: Well done for taking the initiative to perform corrections and justifications in your paper. Your commitment to accuracy and clarity demonstrates strong attention to detail and a dedication to producing high-quality work

**Do you want your identity to be public for this peer review?** For information about this choice, including consent withdrawal, please see our Privacy Policy

Reviewer #1: No

---

## [Editor Report · Acceptance letter]

PONE-D-24-52718R1

PLOS ONE

Dear Dr. Sorjonen,

I'm pleased to inform you that your manuscript has been deemed suitable for publication in PLOS ONE. Congratulations! Your manuscript is now being handed over to our production team.

Kind regards,

on behalf of

Dr. Hikmat Ullah Khan

Academic Editor

PLOS ONE